# Do Protected Areas Matter? A Systematic Review of the Social and Ecological Impacts of the Establishment of Protected Areas

**DOI:** 10.3390/ijerph17197259

**Published:** 2020-10-04

**Authors:** Ben Ma, Yuqian Zhang, Yilei Hou, Yali Wen

**Affiliations:** 1School of Environment and Natural Resources, Renmin University of China, Beijing 100872, China; maben20@ruc.edu.cn; 2Center for Systems Integration and Sustainability, Michigan State University, East Lansing, MI 48823, USA; zhan1364@msu.edu; 3School of Economics and Management, Beijing Forestry University, Beijing 100083, China; houyilei427@163.com

**Keywords:** social effects, ecological effects, protected area, systematic review, conservation mechanism

## Abstract

There is growing interest in evaluating the effects of establishing protected areas (PAs). However, the mechanisms through which the establishment of PAs achieved significant positive effects remain unclear, and how different conservation mechanisms have achieved significant positive social and ecological benefits has also not been sufficiently studied. In this study, we systematically reviewed exemplary cases from Asia, Africa, and South America, using panel data to assess the conservation effectiveness of nature reserves and national parks. By surveying 629 literature samples reported in 31 studies, we found that the establishment of PAs has positive influences on poverty reduction, family incomes, household expenditure, employment, forest cover, biodiversity, carbon sequestration, and a reduction in forest fragmentation. Furthermore, we analyzed the specific aspects that influence the publication of a paper in a high-impact journal. We found that publication is more likely when the research uses panel data, matching methods of data analysis, large samples, and plots or PAs as research units and has significant evaluation results. Our results suggest that future studies should use panel data and matching method analysis to assess the impacts of PAs from multiple perspectives and focus on the effectiveness of specific conservation mechanisms in achieving positive effects.

## 1. Introduction

Establishing protected areas (PAs) is a key strategy for protecting biological resources. The positive effects of PAs on biodiversity conservation are recognized worldwide; however, conservation effectiveness varies considerably [1]. Globally, the land area for conservation has grown tremendously; as of July 2018, the ratio of PAs to total land area has increased to 14.9% [2]. However, conflict exists between conservation and economic development [3] because the establishment of PAs has the potential to affect local livelihoods negatively, especially in developing countries [4,5].

The establishment of PAs has been frequently reported as having negative impacts on local people, such as restricting local residents’ lifestyle (fuel, wood collection, logging, wild plant collection, and livestock grazing) and intensifying human–wildlife conflict (e.g., damage caused by wild animals) [6,7]. However, despite the negative effects, there are significant positive livelihood effects found in many places [8,9] through ecological compensation, reserve-based employment (e.g., as forest rangers or guides), ecotourism revenue sharing, community participation in development projects, and resource access agreements [10,11,12,13].

The primary goal of PAs is to conserve biodiversity, and the significant positive ecological effects associated with them are confirmed by many studies [2,14]. PAs have long been regarded as a critical tool for maintaining habitat integrity and species diversity [15]. Evidence has also emerged that PAs have effectively lessened deforestation in developing countries [16,17,18,19,20,21,22]. For example, approximately 10% of the protected forests of Costa Rica would have disappeared had they not been protected between 1960 and 1997 [16]. However, many of the world’s PAs exist only as “paper parks”, lacking effective management capacity, and, hence, they are unlikely to deliver effective conservation outcomes [23,24].

Poverty is a key factor when assessing the social impacts of PAs. Many studies have examined the relationship between biodiversity conservation policies and poverty [25,26]. Amid strong support for the establishment of PAs based on environmental evidence, poverty advocates are concerned that PAs have the potential to impose unintended negative consequences on local communities [27,28]. Evidence for the establishment of PAs exacerbating poverty has been highlighted in previous studies [5], although many conservation and development professionals do not agree with this depiction [29]. Gurney et al. [30] found that integrated PAs appear to contribute to poverty alleviation. Meanwhile, Andam et al. [17] estimated the effect of PA systems on poverty in Costa Rica and Thailand and found that the net benefit of ecosystem protection reduced poverty. Overall, recent evidence suggests that PAs have not exacerbated poverty [12,17,20,31]; however, studies that estimate the heterogeneity of impacts show that there are socioeconomic winners and losers, depending on geographical locations [8,20,31]. Upton et al. [32] found few significant relationships between indicators of poverty and the extent of PAs at a national scale. 

Many scholars have realized that it may be unsuitable to focus solely on either social or ecological effects because the positive social effects may be at the cost of conservation. To fully understand the effects of PAs, one should consider environmental and socioeconomic outcomes jointly and quantify the heterogeneity in the effect [33] to determine whether ‘‘win-win” scenarios are possible. In this context, Andam et al. [16,17] suggested that Costa Rica’s PA system has reduced deforestation and alleviated poverty. Ferraro et al. [31], however, demonstrated that the environmental and social impacts were spatially heterogeneous. Importantly, the characteristics associated with reduced deforestation are also the characteristics associated with the least poverty alleviation. Miranda et al. [21] assessed whether ‘‘win-win” scenarios were possible in a study of the Peruvian Amazon during the early 2000s and found that PAs reduced deforestation and did not have a significant effect on poverty.

Only by understanding how PAs affect social or ecological outcomes can conservation policies be designed to enhance (mitigate) their positive (negative) impacts [32]. Mechanisms for achieving positive social outcomes have been identified, such as ecotourism, infrastructure construction, and community engagement [34]. Ecotourism in nature reserves has also been widely discussed in the literature regarding its capacity to alleviate poverty in local communities [35]. Ferraro and Hanauer [34] found that ecotourism activities in nature reserves account for approximately two-thirds of the poverty alleviation effect, based on a poverty index that captured residents’ employment status, household appliance ownership, and utility use. Yergeau et al. [36] found that conservation combined with ecotourism is indeed positively related to local welfare. However, ecotourism in PAs is considered to cause some negative social and ecological effects, such as increasing income inequality and natural resource consumption [35]. Oldekop et al. [1] found that PAs that explicitly integrate local people as stakeholders tend to be more effective at achieving positive outcomes in both biological conservation and socioeconomic development.

Therefore, it is not clear how the establishment of PAs delivers positive effects in terms of both social and ecological indicators, and the mechanisms to evaluate the social and ecological outcomes are not yet established. To fill this knowledge gap, we seek to identify how the social and ecological effects of PAs can be evaluated and what mechanisms contribute to positive effects by conducting a systematic review and integrating findings from existing studies. Besides, we also consider the factors affecting the likelihood of research on this topic being published in a high-impact journal and being cited frequently in other works. This is important to understand whether the factors influencing PAs are perceived as having a positive or negative effect. We address this through a systematic review of the empirical literature.

## 2. Data and Methods

### 2.1. Data Collection

Data for this study were collected by searching keywords via the Web of Science database. Keywords used to search for information relating to PAs included “conservation”, “protected areas”, “national park”, and “nature reserve”. Keywords related to social impact included “livelihood”, “poverty”, “income”, and “economic”, and keywords related to ecological impact included “forest cover” and “biodiversity”. Our approach was to select a PA keyword combined with either a social impact or an ecological impact keyword. It is barely possible to provide an exhaustive list of keywords; hence, we attempted to use comprehensive and general keywords. When we found new indicators in any article, they were included as keywords to identify other relevant articles. The search criteria limited the literature selection to only include studies using quantitative research that has been published in international journals with strict peer-review processes. There is a difference between “keywords” and “indicators” in the context. The “keywords” refers to the ones used in the search string (e.g. protected areas, livelihood, forest cover, etc.), while the “indicators” are specific estimates of ecological or social impacts. We reviewed the search results, and the papers using quantitative analysis on the effects of the establishment of PAs were selected as the basis for bibliometric measurement.

We created a literature review summary table to categorize the results with headings that included author information, publication characteristics, research-area characteristics, nature reserve characteristics, sample size, data type, research object, research method, research start and end year, research results characteristics, ecological effect variables, social effect variables, and impact mechanism characteristics. On this basis, the collated documents were entered and categorized. A total of 31 papers closely related to the subject matter were collected, and the results of the literature research were compiled according to the design of the header. A total of 629 records were found relating to the evaluation of the effects of the PAs.

### 2.2. Empirical Strategy

#### 2.2.1. Factors Causing the Evaluation Outcomes

It is important to detect the conservation mechanisms within the literature that help achieve positive and significant results. Thus, in this study, we analyzed the factors causing positive and significant outcomes for conservation. The definition given by the author in an article determines whether the results are significant. Typically, the results are significant at the 10% level.

In this study, the dependent variable, y, can take two different values. y = 1 represents the case in which the evaluation effects are positive and significant, and y = 0 indicates all other results. The independent variables are the publication characteristics, data, method, and conservation mechanism. If we use the linear probability model (LPM), two problems arise. First, the error term is correlated with the independent variables. Second, the dependent takes the value of 0 or 1, but the predictions of y estimated by LPM are greater than 1 or less than 0, which is impractical. Thus, the binary logistic model was applied to address these two issues [37]. The basic form of the model is
(1)$$\rho =F(y=1|X_{i})=\frac{1}{1+e^{-y}}$$
where ρ represents the probability of the evaluation effects being positive and significant, the independent variable *X_i_* represents various influencing factors, and i=1,2,⋯⋯,n. *y* can be expressed as a linear combination of the variables *X_i_*, which is
(2)$$y=\beta_{0}+\beta_{1}X_{1}+\beta_{2}X_{2}+\cdots \cdots +\beta_{n}X_{n}$$
where $\beta_{i}$ refers to the independent variable’s coefficient obtained by the maximum likelihood estimation. Combine (1) and (2), and the Logit model can be expressed as follows:(3)$$\ln\left(\frac{\rho }{1-\rho }\right)=\beta_{0}+\beta_{1}X_{1}+\cdots +\beta_{n}X_{n}+\varepsilon$$
where $\beta_{0}$ represents the constant term and ϵ represents the random error term.

#### 2.2.2. Factors Affecting the Published Paper Influence

Scholars often seek to publish their research papers in high-impact journals to obtain widespread influence. We determined the impact of the research papers through two aspects: journal impact factor and the average number of citations per year. These two aspects combined were used as the dependent variable. Although the explanatory variables may differ, unobservable factors can concurrently influence the journal impact and the average number of citations; therefore, the random disturbance terms are related. Thus, we combined the two equations to estimate the efficiency of the evaluation.

The dependent variable is represented by the journal impact factor and the average citations per year, while evaluation outcome characteristics, research data, research area, research method, conservation mechanism, and social and ecological evaluation indicators are the main independent variables. Hence, we derived the equations as follows:(4)$$Y_{1i}=\alpha_{0}+\alpha_{1}X_{1i}+\varepsilon_{i}$$
(5)$$Y_{2i}=\beta_{0}+\beta_{1}X_{2i}+\mu_{i}$$

In Equations (4) and (5), $Y_{1i}$ and $Y_{2i}$ show the journal impact factor and the average number of citations per year, respectively. $X_{1i}$ and $X_{2i}$ are the influencing factors of the variable vector group, $\alpha_{1}$ and $\beta_{1}$ are the coefficient vectors, and $\varepsilon_{i}$ and $\mu_{i}$ are the stochastic disturbance. 

## 3. Results

### 3.1. Publication Characteristics

Among the 31 journal articles retrieved, the maximum number of articles (four) were published in the *Proceedings of the National Academy of Sciences* and *World Development*, and two were published in the *Philosophical Transactions of the Royal Society B: Biological Sciences*. There were also two papers each in the *Journal of Environmental Economics and Management*, *Environmental Research Letters*, *Environmental and Resource Economics*, *Ecological Economics*, and *Biological Conservation*. All these articles were published in ecology, social sciences, or comprehensive journals and have clear impact factor levels (See Appendix A). The average impact factor of those published journals in 2018 was 5.35, reflecting the fact that the selected articles were published in well-respected high-impact journals. The Google Scholar search engine was used to count the citations of each article. The results showed that the average number of articles cited was 60. The annual average number of citations for each article (calculated as citation/2020-publishing time) was 9.6 by the end of July 2019. We also discovered that the study of the social and ecological effects of PAs began in the 1950s [38]. Quantitative research mainly started to appear after 2000, and most of the studies that met the search criteria of this study were concentrated in 2010 and beyond. The overall related research showed an increasing trend, with 2015 having the most published works fitting the search criteria.

### 3.2. Social and Ecological Evaluation Indicators Selection

The selected research contained 15 papers that evaluate the social effects, accounting for 48.4%, and 10 papers that evaluate the ecological effects, accounting for 32.3%. There were six articles on the comprehensive evaluation of both social and ecological effects, accounting for 19.3%. 

Most of the existing research focused on either the social or ecological effects of PAs but not on the joint assessment of social and ecological effects. This reflects the need for future research to assess the social and ecological impact jointly, which will help to improve the current emphasis on unilateral effects in the policy design process. At the same time, it also reflects the importance of interdisciplinary cooperation in the evaluation of conservation policy effects. Cooperation between economists and ecologists can better estimate the impact of the establishment of PAs.

The data sources, research methods, and research results of 31 articles were collated, and the design indicators were quantified. A total of 629 cases were established to evaluate the impact of PAs, of which 369 cases (58.7%) were evaluated for their social impacts and 260 cases (41.3%) for their ecological impacts. When selecting socially or ecologically relevant evaluation indicators, the following principles were followed: (1) all indicators were converted into positive indicators; (2) some indicators, such as population size and growth, were disregarded as unmeasurable or unusable in the context of this study; (3) when new indicators were found, they were used as keywords to search for more suitable articles. Finally, there were many indicators for evaluating social effects, including poverty reduction, per capita income, wellbeing, consumption levels, reduction in inequality, crop yields, welfare, employment, labor supply, and access to market. Meanwhile, indicators for assessing ecological effects included forest cover (including reduced deforestation and increased forest growth), biodiversity, carbon sequestration, and reduced fragmentation of forested land.

### 3.3. Differences in the Significance of Evaluation Results

There were 450 (72%) cases confirming the positive effects of PAs, of which 336 (75%) positive results were significant (Table 1). There were 179 (28%) negative cases, of which 74 (41%) of the negative results were significant (i.e., *p* < 0.10).

Specifically, in the social impact assessment studies, 59% of the evaluation results were positive; that is, the establishment of PAs had positive social effects. Some 41% of the evaluation results were negative. In the positive social effect evaluation results, 65% of the results were significant, while 35% of the results were not significant. In the negative social effects evaluation, 42% of the results were significant, while 58% of the results were not significant.

In the ecological impact assessment study, 90% of the evaluation results were positive—the establishment of PAs has produced positive ecological effects. Some 10% of the evaluation results were negative. In the positive ecological effect evaluation results, 83% of the positive results were significant, while 17% of the results were not significant. Among the evaluation results showing a negative ecological effect, 37% of the results were significant, while 63% of the results were not significant. This indicates that the establishment of PAs has produced positive effects overall, especially in terms of ecological effects, which has been widely confirmed in empirical research. However, while the results for social effects show more positive cases than negative ones, there are still many significant negative empirical results.

There are many indicators for evaluating social effects, with the largest number of cases using poverty reduction indicators, accounting for 45%, followed by per capita income, employment, and labor indicators, accounting for 18%, 9%, and 9%, respectively (Figure 1). The indicators for evaluating ecological effects were concentrated on forest cover, accounting for 89%, followed by carbon sequestration and forestland fragmentation, accounting for 6% and 5%, respectively (Figure 2). In total, 629 literature samples were analyzed. The indicator frequency measured the number of samples that used specific indicators.

As shown in Table 1, choosing different evaluation indicators affects the evaluation results. In the evaluation of social impact indicators, significant positive cases account for 38%, followed by negative but not significant cases and significant negative cases, reaching 20% and 17%, respectively. Specifically, significant positive cases apply mostly to poverty reduction, per capita income, expenditure, employment, labor supply, and access to market, which indicates that the establishment of PAs has achieved significant positive effects in those aspects. However, in wellbeing, reducing inequality, and welfare, there are more negative cases than positive, indicating that the establishment of PAs has achieved negative effects in those aspects. In the evaluation of ecological impact indicators, 75% of cases are reported to have achieved significant positive effects, and 90% of cases are positive, which indicates that the positive ecological effects of PAs are widely accepted. In all ecological indicators, the proportion of significant positive cases is the highest, especially in biodiversity conservation and forestland fragmentation reduction, with both exceeding 90%. Meanwhile, the proportion of significant negative cases is less than 10% in all ecological indicators. Additionally, the evaluation of ecological effects is focused mainly on forest cover, while the evaluation of social impacts includes many more indicators.

### 3.4. Descriptive Analysis of Main Bibliometric Characteristics

Most of the impact evaluation results are significant and positive (53%), with only 11.9% being significant and negative (Table 2). The social and ecological outcome characteristics were significantly different at the 1% significance level. The positive and significant results of the ecological cases were significantly greater than those of the social cases, while the negative significant results for the ecological outcome cases were significantly less than those in the social cases.

In terms of publication characteristics, the selected literature has a high degree of influence. The impact factor of the published journals in 2018 (from the *Journal Citation Report*, 2019 [39]) reached 5.4, and the average article was cited 11 times per year with an average of 70 citations in total. 

The case studies of social effects were found to be published in journals with lower impact factors than in journals that published ecological effects studies, but the total number of citations per paper on social effects studies was higher than those related to ecological effects studies. There was no significant difference in the number of average citations per year per paper.

When reviewing the characteristics of the journal articles, we found that 55.8% of the cases used panel data, and most were published after 2000. The studies included low to high-income areas, with 49% of the cases in South America, 24.3% in North America, 22.7% in Asia, and 3.97% in Africa, and with an average sample size of 38,550 people. The main topic was national parks for 19.4% of the case studies and nature reserves for 7.8%. For the research unit, 77.58% of the cases used plots, 16.8% used farmers, and 5.56% used PAs. Moreover, there were significant differences between social and ecological cases regarding the data used: most of the relevant studies on the evaluation of ecological effects used panel data, while the research on social effects evaluation mostly used cross-sectional data. The sample size for research regarding ecological effect evaluation was also significantly larger on average than the social effect studies. In terms of the research unit, the research on ecological effects is mostly based on plots, while the research on social effects assessment is mostly based on households.

### 3.5. Analysis of the Influence Mechanisms of PAs

It is important to determine the mechanisms through which the establishment of PAs has achieved significant positive outcomes. This study uses an econometric model to analyze the effect of research data, study area, method, conservation mechanism, and evaluation indicators selection on achieving significant positive outcomes (Table 3).

Compared with other data types, using panel data is more likely to result in significant positive outcomes, especially for social outcomes. The panel data set is more robust and reliable, and it can solve some endogeneity problems, which further confirms the positive effects of PAs. The positive effects of PAs in areas with high economic development levels are more significant than those with low economic development levels both in social and ecological outcomes. Compared with Asia, the establishment of protected areas in South America is more likely to produce positive and significant levels. 

The probability that PAs produce significant positive effects reduces in more recent studies, especially in social outcomes, which reflects increasing conflicts between conservation and development in developing countries. National parks have a lower probability of achieving a significant positive effect compared to other types of protected areas, especially in ecological outcomes. Additionally, the probability of achieving significant positive outcomes by using plots as research units is lower than that of using farmer households as research units, reflecting the fact that the conflicts between communities and PAs are weakening, while macro-level conflicts between regional development and conservation are exacerbating. The matching research method cannot obtain significant positive outcomes as compared to other methods, especially in ecological outcomes, indicating that a regression method, such as ordinary least squares (OLS), will exaggerate the positive significant conclusion. PAs with ecotourism are more likely to achieve significant positive effects than those without. Overall, the evaluation results did not show the same level of positive and significant research results for social indicators as compared to ecological indicators, highlighting that the positive impact of the PAs is mainly reflected in the ecological effects and that there is still room for improvement in social effects.

### 3.6. Analysis of Factors Affecting a Paper’s Potential to Be Published in High-Impact Journals and the Average Number of Citations Per Year

To influence policy development, an article needs to reach a wide, or perhaps respected, audience. Two important indicators for the influence of academic papers are being published in high-impact journals and the number of citations. Because of the correlation between the number of citations and journal impact factor, we use a seemingly uncorrelated regression for joint estimation to improve estimation efficiency.

As shown in Table 4, we found that using significant evaluation results, panel data, large samples, plots or PAs as research units, and the matching method analysis is more likely to increase the probability of being published in a high-impact journal. Meanwhile, attention to ecotourism, econometric methods (such as panel regression), and concerns about poverty reduction and income are usually published in social science journals, which have a lower impact factor than natural sciences.

We also found that the significance, positive or negative, of the results will not affect the average number of citations per year for the paper. However, several factors could increase the number of citations, such as using panel data, paying particular attention to Africa and South America, increasing the sample size, undertaking assessments of the effectiveness of conservation of nature reserves and national parks, adopting the matching method to analyze the ecotourism mechanism, and focusing on the effect of PAs on poverty reduction.

## 4. Discussion

In summary, the results of the literature evaluation confirm the positive ecological effects of PAs. The results demonstrate certain positive social effects but also many negative social effects. The primary goal of PAs is to protect biodiversity, and community natural-resource-use behaviors, such as fuelwood and wild plant collection and logging, are often seen as threats to conservation and are prohibited inside PAs [6]. PAs are controversial, and it is hard to find a win-win solution to conservation and development [28]. Further, it is well recognized that PA management should integrate local development and that the goals of conservation and development should be combined [26].

This study has further confirmed the positive social effects of tourism in PAs, which is consistent with the findings reported by Andam et al. [17], Ferraro et al. [31], Lonn et al. [41], and Ma et al. [42]. However, PAs with tourism have caused negative ecological effects [43]. This shows that tourism is not currently a perfect solution for both biodiversity conservation and poverty alleviation and that there are unintended negative effects to tourism development, such as habitat loss and forest fragmentation. In addition, although payment for ecosystem (PES) is widely applied worldwide, especially in conservation areas [44,45], PES in PAs has not achieved significant positive social effects. It may be necessary to have a trade-off between social and environmental outcomes [46]. Additionally, there may be some unintended negative impacts on local households [7]. Thus, the design of PES in PAs should consider local benefits and the spillover and feedback effects [47,48].

This study has some implications for future studies. First, panel data should be collected and used as much as possible to improve the robustness of research results and to capture the dynamic changes of policy impacts. Second, a matching method analysis, such as propensity score matching or difference in differences, should be adopted to increase the credibility of the research results and reduce endogenous problems. Third, when assessing the impacts of PAs, few studies simultaneously consider both the social and ecological impacts and evaluate the trade-off or win-win outcomes. It is necessary to evaluate the issue from multiple perspectives and consider both the ecological and social indicators to increase the probability of articles being cited and published in high-impact journals. Finally, more attention should be paid to the mediating effects of conservation mechanisms, such as ecotourism and PES, to improve the impact and publication probability of papers.

## 5. Conclusions

In retrieving literature on the social and ecological outcomes of the establishment of PAs, a total of 629 cases were collected. We found this to be a popular research topic in the past decade, with most papers published in prestigious journals and highly cited by many researchers. When evaluating the social impacts of PAs, we found that poverty, income, employment, and labor are the four main indicators used in research; of these, poverty accounts for 45% of the cases. When evaluating the ecological impacts of PAs, most cases focus on forest cover, accounting for 89%. 

The literature reveals that the establishment of PAs has achieved positive effects in terms of poverty reduction, income, expenditure, employment, forest cover, biodiversity, carbon sequestration, and reducing forest fragmentation. In addition, we found some important mechanisms for achieving significant positive conservation effects. For example, significant positive effects are more easily achieved in more economically developed countries, and the establishment of national parks is less effective at achieving significant positive effects than other conservation mechanisms. We found that ecotourism in PAs is an important mechanism for achieving significant positive effects, especially for social effects. For research to have widespread influence, we found that using significant evaluation results, panel data, large samples, plots or PAs as research units, and the matching method result in a higher likelihood of having papers published in a high-impact journal. We also found that using panel data, focusing on Africa and South America, increasing the sample size, assessing the conservation effectiveness of nature reserves and national parks, adopting a matching method of analysis, analyzing the ecotourism mechanism, and addressing the poverty reduction effect were all associated with increasing the number of citations of the research paper. Although the results show a focus of research on certain topics over others, other important factors should also be detected in the future, such as the effect of COVID-19 on conservation and development.

## Figures and Tables

**Figure 1 ijerph-17-07259-f001:**
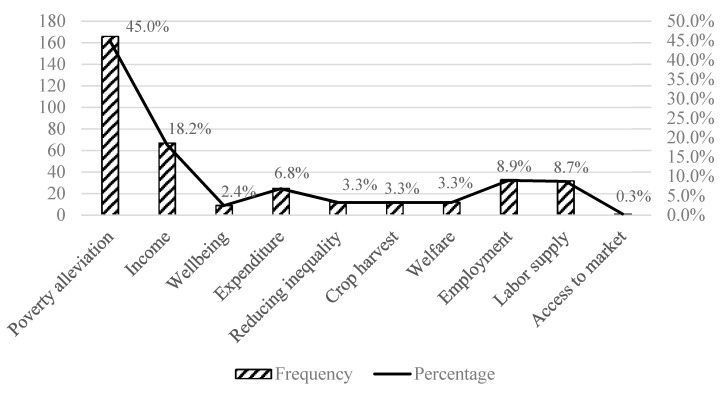
Evaluation indicator distribution of the social effects of protected areas by number and by percentage.

**Figure 2 ijerph-17-07259-f002:**
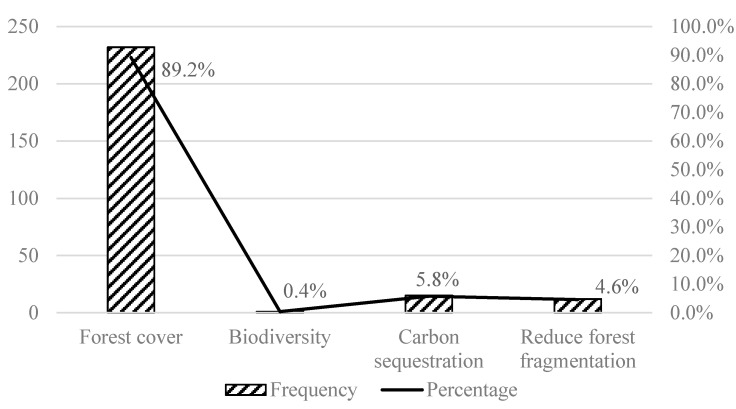
Evaluation indicator distribution of the ecological effects of protected areas by number and by percentage.

**Table 1 ijerph-17-07259-t001:** Descriptive analysis of the evaluation results of the establishment of protected areas.

	Positive			Negative			Total
	Significant	Nonsignificant	Positive in Total	Significant	Nonsignificant	Negative in Total	
**Social indicator**							
Poverty alleviation	70 (42%)	27 (16%)	97 (58%)	22 (13%)	47 (28%)	69 (42%)	166 (45%)
Income	27 (40%)	14 (21%)	41 (61%)	10 (15%)	16 (24%)	26 (39%)	67 (18%)
Wellbeing	3 (33%)	1 (11%)	4 (44%)	4 (44%)	1 (11%)	5 (56%)	9 (3%)
expenditure	10 (40%)	9 (36%)	19 (76%)	0 (0%)	6 (24%)	6 (24%)	25 (7%)
Reducing inequality	0 (0%)	3 (25%)	3 (25%)	2 (17%)	7 (58%)	9 (75%)	12 (3%)
Crop harvest	4 (33%)	7 (58%)	11 (92%)	0 (0%)	1 (8%)	1 (8%)	12 (3%)
Welfare	0 (0%)	4 (33%)	4 (33%)	4 (33%)	4 (33%)	8 (67%)	12 (3%)
Employment	14 (42%)	5 (15%)	19 (58%)	11 (33%)	3 (9%)	14 (42%)	33 (9%)
Labor supply	13 (41%)	5 (16%)	18 (56%)	11 (34%)	3 (9%)	14 (44%)	32 (9%)
Access to market	1 (100%)	0 (0%)	1 (100%)	0 (0%)	0 (0%)	0 (0%)	1 (0%)
Social total	142 (38%)	75 (20%)	217 (59%)	64 (17%)	88 (24%)	83 (41%)	369 (59%)
**Ecological indicator**							
Forest cover	172 (74%)	35 (15%)	207 (89%)	9 (4%)	16 (7%)	25 (11%)	232 (89%)
Biodiversity	1 (100%)	0 (0%)	1 (100%)	0 (0%)	0 (0%)	0 (0%)	1 (0%)
Carbon sequestration	10 (67%)	3 (20%)	13 (87%)	1 (7%)	1 (7%)	2 (13%)	15 (6%)
Reduce forest fragmentation	11 (92%)	1 (8%)	12 (100%)	0 (0%)	0 (0%)	0 (0%)	12 (5%)
Ecological total	194 (75%)	39 (15%)	233 (90%)	10 (4%)	17 (7%)	27 (10%)	260 (41%)
Total	336 (53%)	114 (18%)	450 (72%)	74 (12%)	105 (17%)	179 (28%)	629

**Table 2 ijerph-17-07259-t002:** Descriptive analysis of main variables.

Variables	Explanation	Total (*n* = 629)	Social Outcomes (*n* = 369)	Ecological Outcomes (*n* = 260)	Sig
		Mean ± Std	Mean ± Std	Mean ± Std	
**Outcome characteristic**					
Impact direction	1 = positive, 0 = negative	0.715 ± 0.452	0.588 ± 0.493	0.896 ± 0.306	0.000
Significance	1 = significant, 0 = nonsignificant	0.652 ± 0.477	0.558 ± 0.497	0.785 ± 0.412	0.000
Positive and significant	1 = Yes, 0 = no	0.534 ± 0.499	0.385 ± 0.487	0.746 ± 0.436	0.000
Negative and significant	1 = Yes, 0 = no	0.119 ± 0.324	0.176 ± 0.381	0.038 ± 0.193	0.000
**Publication characteristic**					
Average cites per year		11.5 ± 8.5	11.7 ± 10.3	11.2 ± 4.7	0.462
Journal impact factor in 2018		5.404 ± 2.747	4.991 ± 2.319	5.989 ± 3.173	0.000
Total cites		70.4 ± 81.8	78.4 ± 101.7	58.9 ± 36.3	0.000
**Research data**					
Data type	1 = panel, 0 = others	0.558 ± 0.497	0.366 ± 0.482	0.831 ± 0.376	0.000
Research-area economic development	1 = low income, 2 = lower middle income, 3 = upper middle income, 4 = high income	2.905 ± 1.200	2.829 ± 0.834	3.012 ± 0.378	0.001
Study end year	year	2005 ± 5.828	2006 ± 6.166	2003 ± 4.550	0.000
Samples		38,550 ± 74,287	15,401 ± 18,074	71,405 ± 105,213	0.000
Nature reserve	1 = nature reserve as study object, 0 = others	0.078 ± 0.268	0.089 ± 0.286	0.062 ± 0.241	0.199
National park	1 = national park as study area, 0 = others	0.194 ± 0.396	0.157 ± 0.364	0.246 ± 0.432	0.005
Sample unit	1 = household, 2 = land parcel, 3 = PA	1.887 ± 0.460	1.789 ± 0.565	2.027 ± 0.162	0.000

The classification of research-area economic development is determined according to World Bank Country and Lending Groups [40].

**Table 3 ijerph-17-07259-t003:** Factors affecting the evaluation outcomes of the establishment of protected areas to be significant and positive.

Variables	All Samples	Social Outcomes	Ecological Outcomes
	Coefficient (Std. Error)	Coefficient (Std. Error)	Coefficient (Std. Error)
Data type	2.762 *** (0.590)	2.309 ** (1.033)	0.350 (0.947)
Economic development	1.317 *** (0.240)	1.150 *** (0.317)	1.367 * (0.725)
Africa (Asia as a benchmark)	−0.198 (0.794)	0.222 (1.096)	−1.340 (1.657)
North America	−1.546 *** (0.411)	0.019 (1.234)	−0.737 (0.653)
South America	1.400 *** (0.454)	1.628 ** (0.816)	1.405 * (0.766)
Research end year	−0.0759 ** (0.036)	−0.118 * (0.069)	−0.006 (0.055)
Samples	0.000 (0.000)	−4.33 × 10^−5^ *** (0.000)	−0.000 (0.000)
Nature reserve	−0.371 (0.499)	0.485 (0.867)	0.268 (0.960)
National park	−0.712 * (0.395)	0.445 (0.617)	−1.811 ** (0.853)
Land parcel (household as a benchmark)	−1.960 *** (0.570)	−1.381 * (0.789)	
PA	0.417 (0.886)	0.624 (1.315)	−0.438 (1.637)
Matching	−1.034 * (0.582)	−0.359 (0.825)	−2.934 ** (1.382)
OLS	0.345 (0.595)	0.652 (0.782)	−0.234 (1.293)
Panel data	0.009 (0.692)		−0.109 (1.402)
Tourism as a conservation mechanism	1.694 *** (0.286)	2.033 *** (0.415)	−0.896 (0.781)
Payment for ecosystem as a conservation mechanism	0.212 (0.563)	−0.652 (0.736)	
Indicator selection (1 = social, 0 = ecological)	−1.666 *** (0.400)		
Constant	149.4 ** (72.9)	231.2 * (138.1)	10.6 (110.8)

***, **, and * denote significance at the 1%, 5%, and 10% levels, respectively.

**Table 4 ijerph-17-07259-t004:** Factors affecting a paper’s potential to be published in high-impact journals and the average number of citations per year.

Variables	Journal Impact Factor		Average Citations Per Year	
	Coefficient	Std. Error	Coefficient	Std. Error
**Outcome characteristic**				
Impact direction	0.089	0.095	−0.393	0.551
Significance	0.228 **	0.090	0.470	0.520
**Data**				
Data type	3.940 ***	0.184	6.201 ***	1.070
Research-area economic development	−0.458 ***	0.081	1.460 ***	0.470
Africa (Asia as benchmark) North America South America	3.052 ***	0.269	3.481 **	1.558
	2.699 ***	0.159	0.101	0.923
	−1.232 ***	0.158	2.926 ***	0.919
Study end year	−0.001	0.012	−0.743 ***	0.067
Samples	3.32e × 10^−6^ ***	0.000	2.71 × 10^−5^ ***	0.000
Nature reserve	−0.503 ***	0.186	4.967 ***	1.076
National park	−0.738 ***	0.127	2.262 ***	0.734
Land parcel (household as benchmark) PA	0.684 ***	0.190	−0.353	1.101
	3.805 ***	0.301	9.196 ***	1.742
**Method**				
Matching	3.073 ***	0.188	1.792 *	1.088
OLS	0.782 ***	0.175	−1.281	1.014
Panel regression	−2.801 ***	0.216	−5.182 ***	1.252
**Mechanism**				
Tourism	−0.893 ***	0.104	4.941 ***	0.601
Payment for ecosystem	0.646 ***	0.218	−4.572 ***	1.265
**Indicator selection**				
Poverty	−0.216 *	0.123	1.282 *	0.712
Income	−0.441 **	0.171	−4.684 ***	0.993
Forest cover	−1.895 ***	0.149	−6.955 ***	0.862
Carbon sequestration	0.307	0.293	−1.601	1.700
Constant	5.012	23.3	1492 ***	135.0

***, **, and * denote significance at the 1%, 5%, and 10% levels, respectively.

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
