# Peer review of "Do Protected Areas Matter? A Systematic Review of the Social and Ecological Impacts of the Establishment of Protected Areas"

_ijerph, 2020, doi:10.3390/ijerph17197259_

Round 1
Reviewer 1 Report
Line 36.Line 36. Review the data that mentions ... from 1950 to 2018 ... in the reported appointment Protected planet Report 2018, according to figure 3 is growth of protected areas on land and ocean between 1990 and 2018 and a projected growth to 2020 ...
Line 159-160. Please check next affirmation. We also...most research only qualitative discussions. In citation (37) page 71, various citations are reported on research activities not only qualitative, but also quantitative.
Author Response
Responses to Reviewers’ Comments to
Manuscript IJERPH-947389
Do Protected Areas Matter? A Systematic Review of the Social and Ecological Impacts of the Establishment of Protected Areas
We appreciate the comments from the Editor and reviewers. We have itemized all comments and addressed them individually as follows.
Reviewer 1:
Comment 1: Line 36. Line 36. Review the data that mentions ... from 1950 to 2018 ... in the reported appointment Protected planet Report 2018, according to figure 3 is growth of protected areas on land and ocean between 1990 and 2018 and a projected growth to 2020 ...
Response 1: Thanks for your comments. We have checked the report and have made the relevant changes as follows.
As of July 2018, the ratio of PAs to total land area has increased to 14.9%.
Comment 2: Line 159-160. Please check next affirmation. We also...most research only qualitative discussions. In citation (37) page 71, various citations are reported on research activities not only qualitative, but also quantitative.
Response 2: We agree with your comments and have deleted the confusing affirmation.
Reviewer 2 Report
The manuscript is well written and organized. I recommend a review of some points: - difference and reason for use between keywords and effects found. - Methods is not comprehensive, with methods introduced without any explanation why the types of analysis were used. Additionally, there is no mention on how the significance of the results was calculated.
General comments
Introduction
- In general, it is well written and with a good rationale. However, I am not sure if the statement in L91-93 “it is not clear whether the establishment of PAs delivers positive effects in terms of both social and ecological indicators…” in face of so many studies showing the positive (and negative) aspects of PAs. Maybe the question is not “if” PAs have positive effects but “how” and “when”. I suggest authors review the Introduction in order to be sure if the goals and statements are balanced.
Methods
- There is no explanation why those models were used and their applicability to the topic being analysed. For instance, in L120-123 is said “It is important to detect the conservation mechanisms within the literature that help achieve positive and significant results. Thus, in this study, we analyzed the factors causing positive and significant outcomes for conservation. The binary logistic model was applied. The basic form of the model is:…” This part is very generic and does not provide any insight about its use here. It is necessary to contextualize the use of such models in similar studies and its applicability to the current one.
- Significance of the data are presented in results but not comprehensibly presented in Methods.
Specific comments
L34 – there might be a contradiction between “effects of PAs on biodiversity conservation are recognized worldwide” and L15-17 “whether the establishment of PAs has achieved significant positive effects remains unclear, and how different conservation mechanisms have achieved significant positive social and ecological benefits is also not studied enough”, later on L47-48 and elsewhere. Please check the consistency between those parts.
L48 – please cite some of the many publications that confirm the positive ecological effects of PAs, specially those of broad reach, such as from FAO, etc.
L95 – Capitalize “Pas”.
L101-104 – The keywords are good choices and logical. Although to elaborate an exhaustive list is an elusive task, I miss words such as “food security”, “inequality”, “access to food” for the social aspects, while “ecosystem services”, “habitat integrity”, “fragmentation”, “connectivity” for ecological/environmental aspects.
L178 – Some of the indicators resemble the keywords. Please clarify their differences and why some to the “indicators” were not used as keywords for the search.
L186- Present the results and cite the table. This sentence should be deleted.
L188- please define what is a “significant” effect. It should be described in the methodology, mentioning the possible methods to arrive at that conclusion and how they reflect in this study.
L203 – Delete sentence. Present results and cite the Figure.
L207 – Again keywords and indicators overlap. Please clarify the differences and why some were chosen for keywords.
L210 – Caption is incomplete (Figure 1 and 2). Please add “what” the Figure shows.
Figure 1 and 2 – What is frequency in this context? Please describe in methods (including how it reaches more than 100%).
L232- delete sentence and present data and cite Table.
L241-242 – “journals with lower impact factors than in journals that published ecological effects studies”. It is necessary to note that there are inherent differences between the IF between different fields of knowledge. For example, it would not be fair to compare the IF that cite PAs with those reached to Medicine. As such, ecology and social sciences do have clear IF levels, therefore those characteristics such be addressed.
L352 – authors cite various factors “all help to increase the number of citations of the research paper”. It is not clear if having those topics would increase the number of citations or a higher number of studies on those topics are the reason for the result. Although, it might also show a focus of research on certain topics over others, and in reality, this “bias” could be hindering the understanding of other important factors.

Author Response
Responses to Reviewers’ Comments to
Manuscript IJERPH-947389
Do Protected Areas Matter? A Systematic Review of the Social and Ecological Impacts of the Establishment of Protected Areas
We appreciate the comments from the Editor and reviewers. We have itemized all comments and addressed them individually as follows.
Comment 1: The manuscript is well written and organized. I recommend a review of some points: - difference and reason for use between keywords and effects found. - Methods is not comprehensive, with methods introduced without any explanation why the types of analysis were used. Additionally, there is no mention on how the significance of the results was calculated.
Response 1: Thanks for your comment. We agree and have explained the difference and reason for use between keywords and effects found in the data collection part. In the method part, we explained why the method was used. Also, we explained how the significance of the results was calculated in section 2.2.1.
Comment 2: In general, it is well written and with a good rationale. However, I am not sure if the statement in L91-93 “it is not clear whether the establishment of PAs delivers positive effects in terms of both social and ecological indicators…” in face of so many studies showing the positive (and negative) aspects of PAs. Maybe the question is not “if” PAs have positive effects but “how” and “when”. I suggest authors review the Introduction in order to be sure if the goals and statements are balanced.
Response 2: We agree with you comments and have changed “whether” to “how” to make the statements more appropriate. We also revised the introduction part to make sure the goals and statements are consistent.
Comment 3: There is no explanation why those models were used and their applicability to the topic being analysed. For instance, in L120-123 is said “It is important to detect the conservation mechanisms within the literature that help achieve positive and significant results. Thus, in this study, we analyzed the factors causing positive and significant outcomes for conservation. The binary logistic model was applied. The basic form of the model is:…” This part is very generic and does not provide any insight about its use here. It is necessary to contextualize the use of such models in similar studies and its applicability to the current one.
Response 3: We agree with your comments and have made the relevant changes. we explained why the method was used in section 2.2.1.
Comment 4: Significance of the data are presented in results but not comprehensibly presented in Methods.
Response 4: Thanks for your comments. We have presented how to measure the significance of the data in the empirical strategy part.
Comment 5: L34 – there might be a contradiction between “effects of PAs on biodiversity conservation are recognized worldwide” and L15-17 “whether the establishment of PAs has achieved significant positive effects remains unclear, and how different conservation mechanisms have achieved significant positive social and ecological benefits is also not studied enough”, later on L47-48 and elsewhere. Please check the consistency between those parts.
Response 5: We agree with your comments and have checked the consistency between those parts. We made the relevant changes in the L15-17. It has changed to that through what mechanisms did whether the establishment of PAs has achieved significant positive effects remains unclear.
Comment 6: L48 – please cite some of the many publications that confirm the positive ecological effects of PAs, specially those of broad reach, such as from FAO, etc.
Response 6: We agree with your comments and have citied important publications conducted by FAO and UNEP.
Comment 7: L95 – Capitalize “Pas”.
Response 7: We agree and have made the relevant changes.
Comment 8: L101-104 – The keywords are good choices and logical. Although to elaborate an exhaustive list is an elusive task, I miss words such as “food security”, “inequality”, “access to food” for the social aspects, while “ecosystem services”, “habitat integrity”, “fragmentation”, “connectivity” for ecological/environmental aspects.
Response 8: Thanks for comments and We agree with your suggestions. We made the following explanations. It is barely possible to provide an exhaustive list of keywords; hence, we attempted to use comprehensive and general keywords. When we found new indicators in any article, they were included as keywords to identify other relevant articles. The search criteria limited the literature selection to only include studies using quantitative research that have been published in international journals with strict peer-review processes.
Comment 9: L178 – Some of the indicators resemble the keywords. Please clarify their differences and why some to the “indicators” were not used as keywords for the search.
Response 9: We agree with your suggestions. There is a difference between “keywords” and “indicators” in the context. The “keywords” refers to the ones used in the search string (e.g. protected areas, livelihood, forest cover, etc.), while the “indicators” are specific estimates of ecological or social impacts.
Comment 10: L186- Present the results and cite the table. This sentence should be deleted.
Response 10: We agree and have presented the results and cited the table.
Comment 11: L188- please define what is a “significant” effect. It should be described in the methodology, mentioning the possible methods to arrive at that conclusion and how they reflect in this study.
Response 11: We agree with your comments and have given the definitions in the Section 2.2. The definition given by the author in an article determines whether the results are significant. Typically, the results are significant at the 10% level.
Comment 12: L203 – Delete sentence. Present results and cite the Figure.
Response 12: We agree with your comments and have deleted the sentence, presented results and cited the figure.
Comment 13: L207 – Again keywords and indicators overlap. Please clarify the differences and why some were chosen for keywords.
Response 13: We agree with your comments. We made the following explanations. It’s hard to elaborate an exhaustive list for keywords, so we try to use comprehensive and general keywords. When we find new indicators in the article, they will be included as keywords to find if there are any more relevant articles.
Comment 14: L210 – Caption is incomplete (Figure 1 and 2). Please add “what” the Figure shows.
Response 14: We agree and have made the relevant changes.
Comment 15:Figure 1 and 2 – What is frequency in this context? Please describe in methods (including how it reaches more than 100%).
Response 15: We agree with your comments and have explained the frequency in this context.
629 literature samples were analyzed. The indicator frequency measured the number of samples used this indicator.
Comment 16: L232- delete sentence and present data and cite Table.
Response 16: We agree with your comments and have made the relevant changes.
Comment 17: L241-242 – “journals with lower impact factors than in journals that published ecological effects studies”. It is necessary to note that there are inherent differences between the IF between different fields of knowledge. For example, it would not be fair to compare the IF that cite PAs with those reached to Medicine. As such, ecology and social sciences do have clear IF levels, therefore those characteristics such be addressed.
Response 17: We agree with your comments and have made the following explanations.
All these articles are published in ecology, social sciences or comprehensive journals and have clear impact factor levels (See Supplementary information)
Comment 18: L352 – authors cite various factors “all help to increase the number of citations of the research paper”. It is not clear if having those topics would increase the number of citations or a higher number of studies on those topics are the reason for the result. Although, it might also show a focus of research on certain topics over others, and in reality, this “bias” could be hindering the understanding of other important factors.
Response 18: Great point! we agree with your comments and have addressed the shortcomings of this study and discussed the future research directions.
Reviewer 3 Report
Thank you for this interesting study, I enjoyed reading it. However, you will have to spend some more work on the results and discussion section. Some parapgraphs in the results section are clearly part of the discussion. In addition, you should elaborate more on the interpretation of your results and support this with further references in the discussion section.

Author Response
Responses to Reviewers’ Comments to
Manuscript IJERPH-947389
Do Protected Areas Matter? A Systematic Review of the Social and Ecological Impacts of the Establishment of Protected Areas
We appreciate the comments from the Editor and reviewers. We have itemized all comments and addressed them individually as follows.
Comment 1: Thank you for this interesting study, I enjoyed reading it. However, you will have to spend some more work on the results and discussion section. Some paragraphs in the results section are clearly part of the discussion.
Response 1: We agree with your comments are some paragraphs in the results section has been moved to discussion part.
Comment 2: In addition, you should elaborate more on the interpretation of your results and support this with further references in the discussion section.
Response 2: We agree with your comment and have elaborate more on the interpretation of results and further connected findings to existing literature.